# Patient perspectives on patient similarity-based risk communication for uncontrolled type 2 diabetes in primary care: A qualitative study

**Ruiheng Ong**[1]*, **Chirk Jenn Ng**[1,2], **Kalaipriya Gunasekaran**[1], **Hang Liu**[3], **Wynne Hsu**[3], **Mong Li Lee**[3], **Ngiap Chuan Tan**[1,2]

**1** SingHealth Polyclinics, SingHealth, Singapore, Singapore, **2** Family Medicine Academic Clinical Programme, SingHealth-Duke NUS Academic Medical Centre, Singapore, Singapore, **3** Institute of Data Science, National University of Singapore, Singapore, Singapore

* ong.ruiheng@singhealth.com.sg

## Abstract

### Background

The inertia to adopt protective health behaviours by patients with diabetes is contributed by underestimation of their risks of diabetes complications. Risk communication, using social comparison of glycaemic control and disease trajectory to other patients of similar clinicodemographic profiles, has potential to increase patients' risk perceptions and motivate protective health behaviours. A digital tool named PERDICT.AI was designed to support primary care physicians (PCPs) in patient similarity-based risk communication to patients with type 2 diabetes mellitus (T2DM).

### Aim

This study explored the perspectives of patients with uncontrolled T2DM on how their diabetes-related risks were communicated to them by a PCP using PERDICT.AI.

### Methodology

A qualitative design was used. Eighteen participants aged 40–79 with T2DM with ≥1 HBA1c reading ≥8.0% within the last 6 months were recruited from a primary care clinic in Singapore. Each participant went through the risk communication session followed by an in-depth interview. The transcripts were coded and analysed to identify emerging themes.

### Results

Five themes emerged representing participants' risk perceptions post-session and social comparison tendencies. These are: 1) 'I am myself', 2) motivation to be like the majority, 3) avoiding similar negative outcomes, 4) low risk does not equate to zero

**Data availability statement:** All relevant data are within the manuscript and its Supporting Information files.

**Funding:** This research/project is supported by the National Research Foundation, Singapore under its AI Singapore Programme (AISG Award No: AISG-GC-2019-001-2B). The URL of the funder website is "https://aisingapore.org/jarvisdhl-transforming-chronic-care-for-diabetes-hypertension-and-hyperlipidemia-dhl-with-ai/". The sponsors or funders did not play any role in the study design, data collection and analysis, decision to publish, or preparation of the manuscript.

**Competing interests:** I have read the journal's policy and the authors of this manuscript have the following competing interests: WH, MLL and NCT are Principal Investigator and Co-Principal Investigators respectively of "JARVIS-DHL: Transforming Chronic Care for Diabetes, Hypertension and HyperLipidemia (DHL) with AI". This research/project is supported by the National Research Foundation, Singapore under its AI Singapore Programme (AISG Award No: AISG-GC-2019-001-2B). This does not alter our adherence to PLOS ONE policies on sharing data and materials.

risk, and 5) motivation to replicate how others achieve positive outcomes. Themes 1 and 4 were concurrently represented among some participants; zero risk tolerance for diabetes complications was expressed despite not being motivated by their HBA1c cohort ranking.

## Conclusion

This study provided insights into the acceptability of using a social comparison approach in communicating risk to patients with uncontrolled T2DM. It highlights the importance of identifying and selecting patients who are receptive to social comparison, clarifying patients' perceptions of risks, including zero risk, and providing tailored and socially comparable strategies to mitigate these risks.

---

## Introduction

Diabetes carries a huge economic burden in Singapore and around the world [1–3]. Among the diabetic population in Singapore, younger adults tend to have poorer glycaemic control and medication adherence [4–6]. The inertia to adopt protective health behaviours is contributed by knowledge gaps about diabetes and its complications, resulting in underestimation of the risks of diabetes complications [7–11]. To drive intention and uptake of protective health behaviours, it is pertinent that patients with diabetes have accurate risk perceptions about their health conditions [12–14].

A potential strategy to increase risk perception and motivate the adoption of protective health behaviours is to leverage on social comparison for risk communication [15–17], by means of comparing a diabetic patient's glycaemic control and disease trajectory to other patients of similar clinical and demographic profiles [18–22]. Patient similarity-based models are shown to more effectively predict chronic disease outcomes compared to general population-based models [23–27], underpinned by the concept that "similar patients with similar features have similar outcomes" [25]. This is especially prevalent in cancer research where genetic or molecular profiles are used to predict the effectiveness of cancer treatments [25,27]. In the context of diabetes however, patient similarity-based models tend to focus on predicting diabetes onset [24,26] or effectiveness of pharmacological treatment [23], rather than predicting the complications arising from diabetes.

PERDICT.AI (Personalised Diabetes Counselling Tool using Artificial Intelligence), a digital tool based on an AI-driven patient similarity model, was developed by a team of primary care physicians (PCPs) and data scientists to support PCPs in patient similarity-based risk communication during their consultations with patients who have type 2 diabetes mellitus (T2DM) [28,29]. Based on the pilot study involving PCPs [30], risk communication leveraging on PERDICT.AI is likely to be more effective for patients with uncontrolled T2DM and who are receptive to social comparison. Furthermore, PCPs alluded to patients being discouraged or distressed when their glycaemic control is compared to that of other patients, particularly when their glycaemic control is due to circumstances out of their control. This is an unintended

consequence of risk communication about diabetes complications that could interfere with diabetes self-management [31]. The findings informed us to narrow down the target recipients of the risk communication to patients with uncontrolled T2DM, instead of all T2DM patients regardless of glycaemic control. Therefore, this study explored the perspectives and experiences of this patient subgroup on how their diabetes-related risks were communicated to them by a PCP using PERDICT.AI. The findings will allow comparison of perspectives from these two major stakeholders on patient similarity-based risk communication using PERDICT.AI.

## Methodology

### Study design and setting

A qualitative methodology was chosen to explore how patients responded to the risk information PCPs provided to them using PERDICT.AI. This included the complexities and nuances of social comparison influencing patient perceptions of their diabetes and related risks and their motivation in improving their glycaemic control [32]. In-depth interviews were conducted between August 2023 and December 2023 at a public primary care clinic (Polyclinic) within a healthcare institution in the Eastern region of Singapore providing ambulatory primary care to over 200,000 patients with T2DM. The study was part of a larger two-arm pilot randomised controlled trial (RCT) to evaluate the feasibility of delivering the risk communication session using PERDICT.AI in primary care for patients with suboptimal T2DM control.

Ethics approval was obtained from the SingHealth Centralised Institution Review Board (CIRB 2023/2306). Informed written consent was obtained from all participants prior to participation in the study. Reporting of the study was guided by the consolidated criteria for reporting qualitative studies (COREQ).

### Study team and reflexivity

The core study team comprised three PCPs (RO, CJN, NCT) who are practising Family Physicians in ambulatory primary care, and one Research Associate (KG) who formerly practised as a public health physician. RO, KG, CJN and NCT are trained in qualitative research, while CJN and NCT are experienced primary care researchers and hold professorial appointments in Family Medicine. RO, CJN, TNC and KG developed the patient similarity-based risk communication session with additional support from three computer scientists (HL, WH, MLL) who designed the PERDICT.AI tool together with NCT.

### Participants and sampling

Participants were patients aged 40–79 years with T2DM and at least one HBA1c reading ≥8.0% within the last 6 months, proficient in English language, and on follow-up at the study site for at least 12 months. A sample size of 20 was established based on the recommended sample size of 12 per arm by Julious for feasibility studies without prior information to base the sample size on, accounting for a 50% attrition rate at follow up and missing data [33]. For this qualitative study, the sample size was guided by Hennink and Kaiser who found that saturation is often achieved within 9–17 interviews [34].

With permission from the primary care team, potential participants were pre-screened for eligibility using a list generated from the electronic medical records by the data team. Potential participants were recruited by convenience sampling. They were referred from the primary care team or approached by the study team for face-to-face recruitment during their scheduled clinic visit. None of the participants had a prior relationship with the study team. Assurance of data confidentiality was given to all participants. Of the 93 potential participants approached, 40 consented to participate in the study and 53 declined to participate, with 8 citing unavailability of time or inability to commit, 1 citing inconvenience to participate and 44 citing not being keen or interested in the study. Participants were randomised in a 1:1 ratio to attend the risk communication session and interview (intervention arm, N = 20) or receive usual care (control arm, N = 20) in an open-label fashion. Recruitment was done between 21 Aug 2023 and 25 Oct 2023.

## Data collection

Data collection was conducted face-to-face in a quiet room within the study site. A flow diagram of the data collection is illustrated in Fig 1.

**Baseline details.** Upon recruitment, a standardised questionnaire was used to record the baseline demographic details and medical history of the participants.

**Patient similarity-based risk communication session and in-depth interview.** 3-4 weeks after recruitment, participants in the intervention arm attended the patient similarity-based risk communication session (S1 Appendix). Based on Social Comparison Theory (SCT) and Health Belief Model (HBM) [13,15–17] and guided by information from PERDICT.AI (S2 Appendix), a trained study team member who is also a PCP (RO) explained the severity of the participant's HBA1c based on its ranking among a similar-patient cohort (SPC) and the consequences of uncontrolled diabetes based on the SPC's actual complication prevalence rates. Higher HBA1c levels from poorer diabetes control results in a lower rank among the SPC, and is likely to place the patient among the cohort minority who were more likely to develop diabetes complications. Where made available by PERDICI.AI, a positive and negative case example from the SPC was illustrated. The SPC, comprising actual T2DM patients with similar clinical and demographic profiles from the same primary care institution, was used as a basis for social comparison to raise awareness about the seriousness of the participant's diabetes state and the possible illness trajectories following ahead. Pharmacological (e.g., medication adjustments) and non-pharmacological (e.g., diabetes self-care activities) measures were recommended to improve HBA1c. Pertinent observations about participants' engagement during the sessions were recorded in the field notes by another study team member (KG).

An in-depth interview (S3 Appendix) was conducted immediately after the patient similarity-based risk communication session to gather participants' perspectives on the utility of the session. In total, the session and interview took between 38–72 minutes to complete for each participant. The session and interview were both audio-recorded. The first interview was completed on 31 Aug 2023 and the last interview was completed on 4 Dec 2023.

## Data analysis

All audio recordings were transcribed verbatim and checked for completeness and accuracy. RO and KG familiarised themselves with the transcripts from the first two in-depth interviews and independently coded the transcripts. Open codes were assigned to the transcripts based on the study objective, and were combined to form a coding framework. RO and KG met to discuss and reach a consensus on shared meanings of the codes, and discrepancies were resolved through consultation with an experienced qualitative researcher CJN. RO coded the remaining transcripts and new codes were added iteratively after discussion with the study team. The data was managed using NVivo Windows Release 1.5.1. Emergent themes were identified. Thematic saturation was reached after 12 interviews. No new themes emerged from the remaining interviews.

**Fig 1. Flow diagram of data collection.**

## Results

### Participant characteristics

18 of the 20 participants in the intervention arm completed the patient similarity-based risk communication session and interview. Among the 2 participants who dropped out of before the session, 1 cited loss of interest in the study and 1 cited ill health after unsuccessful rescheduling of the session. Characteristics of participants are detailed in Table 1.

**Table 1. Characteristics of participants (N = 18).**

| Participant characteristic | n (%) |
|---|---|
| **Sex** | |
| Male | 8 (44.4) |
| Female | 10 (55.6) |
| **Race** | |
| Chinese | 10 (55.6) |
| Indian | 1 (5.6) |
| Malay | 7 (38.9) |
| **Age (years)** | |
| 40–49 | 2 (11.1) |
| 50–64 | 9 (50) |
| 65–79 | 7 (38.9) |
| **Duration of diabetes (years)** | |
| ≤ 5 | 3 (16.7) |
| 6-10 | 6 (33.3) |
| ≥ 11 | 9 (50) |
| **HBA1c at enrolment (%)** | |
| < 8.0 | 1 (5.6) |
| 8.0-8.9 | 7 (38.9) |
| 9.0-9.9 | 5 (27.8) |
| ≥ 10.0 | 5 (27.8) |
| **Comorbidities** | |
| Hyperlipidaemia | 16 (88.9) |
| Hypertension | 15 (83.3) |
| Cardiovascular disease | 2 (11.1) |
| Chronic kidney disease | 12 (66.7) |
| Eye complications | 9 (50) |
| **Education level** | |
| Secondary school and below | 11 (61.1) |
| Vocational | 2 (11.1) |
| Diploma or Pre-university | 3 (16.7) |
| University degree and above | 2 (11.1) |
| **Smoking status** | |
| Non-smoker or ex-smoker | 17 (94.4) |
| Current smoker | 1 (5.6) |
| **Participant characteristic** | **Mean (σ)** |
| Age (years) | 62.2 (10.6) |
| Duration of diabetes (years) | 10.4 (5.7) |
| HBA1c at enrolment (%) | 9.4 (1.4) |

## Principal findings

Five major themes emerged from the data (Table 2).

**Theme 1: 'I am myself'.** Several participants held the belief that 'I am myself'. They were not concerned about their HBA1c rank among the SPC. Instead, they were motivated by wanting to achieve their personal HBA1c targets and preferred the conventional approach of using individualised targets to frame risk messages on disease severity.

*"I don't bother comparing myself to others. I'll just see how I should manage [my diabetes]… [It is] a bit silly to do that [comparing to others]. Why make myself so stressed up?" (Participant 6)*

*"No need [to compare to others], because 'own self', [the] health [of my 'own self' is] different from other people." (Participant 13)*

*"I have a mindset saying, 'I [am] myself. Why should you want to compare me with others?' This is just a study… you randomly take hundred people, so you may find [that you fall] under this one [group of people] and [because] this is just a study, you can take it or leave it… At the end of the day, it's yourself… All these figures… [whether] true or not, it doesn't matter… People have ego. I [am] myself, why [do] you want to compare me with it [the figures]?" (Participant 14)*

*"Don't compare with other people… I don't care about other people. I care about my own self. I cannot worry about them… [and] I won't compare them. I try my best to take care [of] myself and how to get my diabetes down." (Participant 15)*

*"You cannot compare your health and my health. Different bodies have different kinds [of standards]… I'm not new to this sickness… We try our best… but what can we do right? You lead your normal life, I lead my life. It's different. It's very difficult for us to maintain like what you [the doctor] want us to be... Maybe their [patients with better HBA1c] lifestyles are more stable in everything [and] they don't have family members with the same sickness?" (Participant 17)*

One participant highlighted that controlling diabetes is an individual battle. Despite being among the cohort majority with better diabetes control, it does not equate to being immune from complications.

**Table 2. Emergent themes and subthemes.**

| Themes | Subthemes |
|---|---|
| 1. 'I am myself' | Belief that 'I am myself' |
| | Not concerned about HBA1c rank but motivated by personal HBA1c target |
| | Being in the majority with better diabetes control does not equate to immunity from complications |
| 2. Motivation to be like the majority | Fear of remaining like the minority with lower-ranked HBA1c |
| | Motivated to improve HBA1c to be like the majority with better HBA1c |
| 3. Avoiding similar negative outcomes | Avoiding being like patients whose HBA1c remained high and developed higher complication rates |
| | Wanted to delay onset of complications |
| 4. Low risk does not equate to zero risk | Aversity to any risk of developing complications |
| | Wanted zero risk |
| | Small percentage concerning as this translates to a sizeable number affected |
| 5. Motivation to replicate how others achieve positive outcomes | Motivated to replicate the methods leading to positive outcomes such as improved HBA1c |

*"It is your own battle. It is your own self. In my opinion, being in that peer group [with lower HBA1c] doesn't mean that you are out of the woods… I don't want to be a part of the statistics if I can." (Participant 7)*

One participant expressed interest in knowing the HBA1c trend to know how well the diabetes control has fared over time.

*"I think it is good [if] the doctor shows you that kind of graph and tells you, 'You started with this. Six months ago your [reading] was like that, then now [it] is [like this].' It'll be good if the doctor shows me a graph like that to monitor my trend… to look at what others have and even compare in terms of longitudinal… When you look at it… the trend in the last two, three years. Did it maintain? Did it increase? Did it decrease? It'll be interesting for me." (Participant 3)*

**Theme 2: Motivation to be like the majority.** Upon learning that they were like the cohort minority with lower-ranked HBA1c, several participants expressed fear of remaining in their current state and were motivated to improve the HBA1c to be like the cohort majority.

*"I feel alarmed that mine [HBA1c] is so high compared to the others… Seeing that I belong to the lowest [HBA1c rank], I must make a big effort to lower my [HBA1c]." (Participant 11)*

*"I feel that I can be better. If I can be among them [the majority with good HBA1c levels], why not right? I want to make myself to be [among the] eighty-seven percent [who are] better than me… I want to fall within that group. I don't want to be in the bottom… I'm scared… [Currently] I'm that one [among] the thirteen [percent of] patients. That is me." (Participant 12)*

*"It's like when you're jogging with a hundred people [and] now you're at the back of the twenty-two, definitely you [would] want to be among the seventy-eight… How to get to the seventy-eight? Definitely you must have the motivation or some pushing factor." (Participant 14)*

*"Comparison is good because it helps you to work better. Then, you will tell yourself that if people around the same age can do it, you also can do it. It's like a boost that [improving your HBA1c] can be done… It can motivate me to do better." (Participant 16)*

One participant wanted to improve the HBA1c to be like the middle group.

*"The position [of my HBA1c] where you're in among the group. Let's say out of ten [people], mine is the bottom three. That's the best [most helpful] part… I don't want to be in the bottom. I want to be in the middle." (Participant 17)*

**Theme 3: Avoiding similar negative outcomes.** Several participants expressed concern about developing diabetes complications. They wanted to avoid being like the patients whose HBA1c remained high and developed higher complication rates and this was a wake-up call to improve their diabetes control. Conversely, they aspired to be like the patients whose HBA1c improved over time and developed lower complication rates.

*"If I'm in the red zone it'll be more realistic than seeing a percentage [of the complication rate]… This is the bigger part that helps me to think, 'Oh no, I should be doing something. I should not be there in that zig-zag red [line].'… [With] the fluctuations [in HBA1c levels]… seeing things like that… helps in a way that makes me [think], 'I shouldn't be in that red zone. I should be in the green zone.'" (Participant 3)*

*"[I] must take control [of my diabetes] already, if not I will fall into that category [of diabetic patients with higher complication rates]… if I don't control [my diabetes], I'll end up like them… I don't want to fall into that category."* (Participant 8)

*"[One] should be more careful. You don't join the category [of diabetic patients with higher complication rates], [instead] you should put yourself more onto the [category with] lesser risk… We should be keeping [towards] the safe side [and being within] the comfort zone… I should, as you said, set my goal on the lower side. It only brings good rather than [harm] you know, [rather than] the other way around?"* (Participant 17)

One participant felt that it would be a good outcome if the onset of diabetes complications is delayed.

*"Scary… I don't want to be affected health wise by all these [complications]. If I can take preventative action, I would try and do it. If I don't take care then the risk [of getting these complications] will increase, so that's why I don't want to come to a point where it'll lead to blindness and all those kinds of things… As I said, I want to die healthy… But [if] I can at least delay [the complications by] ten years, [it will be] already good enough. So if I act earlier… I think I can reduce [or] delay the start of these types of complications."* (Participant 6)

**Theme 4: Low risk does not equate to zero risk.** Several participants expressed zero risk tolerance for diabetes complications. They were highly averse to any risk of developing diabetes complications and wanted their risk to be zero. A low prevalence rate of 1% was of significant concern to them, as these participants felt they could be the 1% or that 1% translates to sizeable numbers affected.

*"The higher the percentage of the risk, the more alarmed it makes me… Even [if there is] a slight risk, I'll also be worried… if one percent [of people can] get [these complications], I might be the one percent. Who knows?"* (Participant 11)

*"Definitely there's a chance of getting some complications… Even [if] the [percentage] is one percent, there's a tendency that you may fall into that category… Everybody of course hopes that they are at zero percent… we have to put ourselves into the worse [possible] scenario… never say never… anything can happen… One percent… if you tell me there's ten thousand people, the one percent is… around hundred people. Who says that you will not land among that one hundred people?"* (Participant 14)

*"I still prefer zero percent. That's the best thing. [Although] this three percent is at a very low percentage, inside my heart I think, 'Zero percent is still the best.' Because [as] your age advances, I told you the risk gets higher. Like I said, hopefully because we are just fifty plus, hopefully it's zero percent."* (Participant 16)

**Theme 5: Motivation to replicate how others achieve positive outcomes.** Several participants attributed the example patients' success in HBA1c improvement to medication or lifestyle measures. They were motivated to replicate the methods leading to these positive outcomes, in hopes of improving their own HBA1c. Having detailed information on these methods, where available, would be useful for these participants.

*"If other people can do it, like control their diabetes, it will motivate me more to know that this thing [diabetes] can be controlled… I have to try the plan to see whether it can work. If it works and it's beneficial to me, then obviously it is beneficial to the others."* (Participant 1)

*"Can compare [to other peoples' diabetes] but… how [did] they control it? My problem is maybe I haven't come to that stage [where I know] how to control my diabetes… I'm just wondering why they can do [so while] I cannot… [and] why they can get the results [to be this] good."* (Participant 10)

*"I hope that I can also achieve this type of [HBA1c] reading. I don't think he [the patient example] actually [did this] just by taking medicine… definitely he went through some adjustment to his lifestyle… that's why he can actually get this type of six-point-something [HBA1c]." (Participant 14)*

*"To me, it's good that we [people] share what is happening to them. Even though we are living different lifestyles [and] eating styles, whatever styles we have, it's something good [to share]... It's something that I can learn from you… No big deal about it… if we bring all these things [information] together." (Participant 17)*

*"Is it because I'm among the number of those people [in the minority], [that] actually I fall [among] the worst [HbA1c]? But again, all those people here… are they on the same milligram [of] metformin or higher?" (Participant 18)*

## Discussion

The study revealed diverse risk perceptions among the patient participants. Several of them were motivated to improve their glycaemic control to be compatible with most local patients with T2DM. In contrast, a few were resigned to be among the minority at risks of complications. The findings are consistent with PCPs' perspectives from the previous study that risk communication using PERDICT.AI would more likely benefit patients who favour social comparison. While PCPs in the previous study had perceived that patients could potentially misinterpret or misunderstood their risks in using the PERDICT.AI [30], none of the patients in the present study alluded to such risk when they were compared with other "similar" patients.

With fear and risk aversity elicited among some participants, the study team recognises the potential for patient similarity-based risk communication to cause increased anxiety, frustration, and possibly disengagement in certain patients. This is consistent with the concerns PCPs raised in the previous study [30] and is a known unintended consequence of risk communication about diabetes complications that could interfere with diabetes self-management [31]. These occurrences can be minimised by PCPs' awareness of their patients' contextual factors contributing to glycaemic control, anticipation of negative reactions, and ability to frame and communicate risk information in a person-centric manner while taking the opportunity to motivate patients to improve their glycaemic control.

The findings are consistent with known literature on upward social comparison's ability to engage competitiveness and facilitate information seeking to improve one's health [18]. They are also consistent with known literature on risk perceptions having both "deliberative" (rule-based using absolute or comparative numerical information) and "affective" contextual factors that drive behavioural decisions [14,35–39], as demonstrated by participants who were motivated to reduce their complication risk to zero due to fear of being like the cohort minority who developed complications (i.e., "affective" risk perception), despite the minority being numerically small (i.e., not applying numerical principals seen in "deliberative" risk perception) [40].

Some participants wanted to know how to improve their glycaemic control and lower their risk of diabetes complications, seeking to replicate methods which worked for other patients. Future versions of PERDICT.AI should consider incorporating data on lifestyle measures. Patient narratives, through sharing of personal stories, can be used in tandem to motivate uptake of protective health behaviours and reduce fears among similar patients [41].

Patient similarity-based risk communication should be targeted at patients who are receptive to social comparison. Patients without such inclination may benefit from other modes of counselling to raise their risk awareness. Nuances in social comparison practices among the participants shows that self-reported information on social comparison tendencies alone do not reliably predict whether a patient would be receptive to patient similarity-based risk communication. Therefore, challenges are anticipated in identifying and stratifying the nuanced layers of social comparison practices among patients using existing questionnaires such as INCOM (Iowa-Netherlands Comparison Orientation Measure) [42]. Further research is needed to operationalise the stratification of T2DM patients based on their social comparison tendencies and to develop validated measurement tools for data collection. For example, a potential direction is to prospectively study social comparison occurrences, characteristics, and its consequences on T2DM patients practising behavioural change

using 'ecological momentary assessment' after participants are trained to recognise day-to-day social comparison occurrences. This method intensely captures the nuances of time-sensitive within-person social comparison occurrences and effects over multiple short time periods throughout the day and is less prone to recall bias compared to retrospective self-reporting methods [43]. The findings would shed new insight on how to identify patients who are most receptive to patient similarity-based risk communication and to prioritise its use for these patients, with the aim of advancing their risk awareness and motivating the adoption of protective health behaviours [44]. The effectiveness of the risk communication would also depend on the PCP's ability during a doctor-patient dialogue to frame the information in a person-centered manner, while avoiding the tendency to "climb probability trees" [45,46].

The study has limitations. Convenience sampling, rather than purposive sampling, was used due to challenges in recruiting patients into this study, introducing selection bias and limiting generalisability. Relevant attributes such as ethnicity and health literacy may not be adequately represented. Nevertheless, there was a good representation of the participants in terms of age, gender, education level, diabetes duration and control, which provided some degree of variation. The complication prevalence among the SPC should not be interpreted as an individual's predicted risk of developing complications. For such predictions, validated risk calculators should be used, such as the UK Prospective Diabetes Study (UKPDS) [47] in the UK and Framingham Adult Treatment Panel ATP III [48] in Singapore for cardiovascular disease risk calculation.

## Conclusion

This study provided insights into the acceptability of using a social comparison approach in communicating risk to patients with uncontrolled T2DM. It highlights the importance of identifying and selecting patients who are receptive to social comparison, clarifying patients' perceptions of risks, including zero risk, and providing tailored and socially comparable strategies to mitigate these risks.

## Supporting information

**S1 Appendix. Patient similarity-based risk communication session.**
(PDF)

**S2 Appendix. PERDICT. AI digital tool.** This is a detailed description of the modules in the PERDICT.AI digital tool.
(PDF)

**S3 Appendix. Interview topic guide.**
(PDF)

## Acknowledgments

The authors would like to acknowledge Qiao Gao and Wei Ying Tan from the Institute of Data Science, National University of Singapore for their work in cleaning the dataset and developing the patient similarity algorithm for PERDICT.AI, as well as Patricia T Kin, Yang Thong Tan, Usha Sankari, Paulpandi Muthulakshmi, Jullina Binte Buang, Eileen Yi Ling Koh, Wai Keong Aau, Subramanian Reena Chandhini and Fei Yang Tan from the Department of Research, SingHealth Polyclinics for their support in making this work possible.

## Author contributions

**Conceptualization:** Ruiheng Ong, Chirk Jenn Ng, Kalaipriya Gunasekaran, Hang Liu, Wynne Hsu, Mong Li Lee, Ngiap Chuan Tan.

**Data curation:** Ruiheng Ong, Kalaipriya Gunasekaran.

**Formal analysis:** Ruiheng Ong, Chirk Jenn Ng, Kalaipriya Gunasekaran.

**Funding acquisition:** Wynne Hsu, Mong Li Lee, Ngiap Chuan Tan.

**Investigation:** Ruiheng Ong, Chirk Jenn Ng, Kalaipriya Gunasekaran, Ngiap Chuan Tan.

**Methodology:** Ruiheng Ong, Chirk Jenn Ng, Kalaipriya Gunasekaran, Ngiap Chuan Tan.

**Resources:** Ngiap Chuan Tan.

**Software:** Hang Liu, Wynne Hsu, Mong Li Lee.

**Supervision:** Chirk Jenn Ng, Wynne Hsu, Mong Li Lee, Ngiap Chuan Tan.

**Writing – original draft:** Ruiheng Ong.

**Writing – review & editing:** Ruiheng Ong, Chirk Jenn Ng, Ngiap Chuan Tan.

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
