## [Decision Letter · Decision Letter 0]

PONE-D-24-37711Utility of patient similarity-based risk communication during primary care consultations: Perspectives from patients with uncontrolled type-2 diabetesPLOS ONE

Dear Dr. Ong,

Thank you for submitting your manuscript to PLOS ONE. After careful consideration, we feel that it has merit but does not fully meet PLOS ONE’s publication criteria as it currently stands. Therefore, we invite you to submit a revised version of the manuscript that addresses the points raised during the review process.

We look forward to receiving your revised manuscript.

Kind regards,

Naeem Mubarak, PhD

Academic Editor

PLOS ONE

Journal Requirements:

Additional Editor Comments:

The manuscript has a good deal of merit for publication after minor revisions.

Given the similarities between the two articles, we request the authors to provide a clear justification for the need to present these findings separately. Alternatively, they may consider combining their results into a single, comprehensive manuscript. This approach could strengthen the impact of their study by providing a more holistic perspective on the research question.

Comments from PLOS Editorial Office: We note that one or more reviewers has recommended that you cite specific previously published works. As always, we recommend that you please review and evaluate the requested works to determine whether they are relevant and should be cited. It is not a requirement to cite these works. We appreciate your attention to this request.

Reviewers' comments:

Reviewer's Responses to Questions

**Comments to the Author**

1. Is the manuscript technically sound, and do the data support the conclusions?

Reviewer #1: Yes

Reviewer #2: Yes

Reviewer #3: Yes

2. Has the statistical analysis been performed appropriately and rigorously? 

Reviewer #1: Yes

Reviewer #2: Yes

Reviewer #3: Yes

3. Have the authors made all data underlying the findings in their manuscript fully available?

Reviewer #1: Yes

Reviewer #2: Yes

Reviewer #3: Yes

4. Is the manuscript presented in an intelligible fashion and written in standard English?

Reviewer #1: Yes

Reviewer #2: Yes

Reviewer #3: Yes

5. Review Comments to the Author

Reviewer #1: Utility of patient similarity-based risk communication during primary care consultations: Perspectives from patients with uncontrolled type-2 diabetes

Title: Appropriate and informative.

Abstract: Appropriate and informative.

Number of words: 320 words (Maximum 300 words; not correct according to the instructions of the authors).

Keywords: Please use/add specific MeSH words like; Qualitative analysis, Type 2 diabetes mellitus, Singapore, PERDICT.AI.

Introduction: Appropriate and informative.

Objectives: Appropriate and informative.

Methods:

The title of the section could be changed to be “Methodology” OR “Subjects and Methods” OR “Patients and Methods”.

Appropriate and informative.

Statistical Analysis: Appropriate and informative; needs more details.

Results:

Text of the results: Appropriate and informative; but long.

Please do not repeat in the text what has been presented in the tables.

Discussion: Appropriate and informative, however it needs to discuss with more details every finding of the study.

The Limitations Section of the study; more possible biases?

Conclusion: Conclusions need to be more concise and specific.

References:

Most references are old (>50% of the study references were published before 2019).

Please use Vancouver style in all references.

S1 Table: It is repeated.

Reviewer #2: The document is well written with only one typographical error identified

in the data analysis section which should read: 'RO, KG and CJN met 'to reach' a consensus on the coding framework. instead of 'RO, KG and CJN met 'to reached' a consensus on the coding framework'.

Reviewer #3: Respected Editor,

Thank you for the opportunity to review the manuscript "Utility of Patient Similarity-Based Risk Communication During Primary Care Consultations: Perspectives from Patients with Uncontrolled Type-2 Diabetes." This study explores how patient similarity-based risk communication is perceived by individuals with uncontrolled type 2 diabetes in primary care settings. The research assesses its feasibility and potential impact on patient engagement, decision-making, and diabetes management.

I recommend the following major revisions in the manuscript:

1. The title is informative but making it more specific would help highlight the qualitative nature of the study. The term "utility" is somewhat vague; specifying whether it refers to feasibility, acceptability, or effectiveness could make it clearer. It would be helpful to revise the title to something like “Patient Perspectives on Patient Similarity-Based Risk Communication for Uncontrolled Type 2 Diabetes: A Qualitative Study” to explicitly reflect the study design.

2. In the introduction, the background is well-structured, but it primarily focuses on the rationale for risk communication. While this is important, it would be beneficial to also discuss the possible limitations of social comparison-based approaches. Addressing potential drawbacks, such as patient distress from unfavourable comparisons or unintended negative effects on motivation, would provide a more balanced perspective.

3. The introduction does not clearly state what is already known about patient similarity-based risk communication and the specific gaps this study aims to fill. It would strengthen the introduction to explicitly highlight what makes this study novel by contrasting it with previous research.

4. In the methods, the study design is qualitative, but the justification for choosing qualitative methods is not provided. It would be helpful to explain why this approach was preferred over a mixed-methods design. Clearly stating why qualitative interviews were the most suitable method would improve the clarity of the methodology.

You may consider citing the following study:

DOI: 10.2147/RMHP.S296113

This study provides a conceptual framework and uses qualitative methods to explore collaborative medication therapy management. It can help justify why qualitative research is an appropriate approach in healthcare communication studies.

(This is optional and should only be taken as a suggestion for the improvement of the manuscript.)

5. In the methods, convenience sampling was used, which introduces selection bias. However, this limitation is not explicitly discussed. Acknowledging the potential biases associated with this sampling strategy and discussing its impact on generalizability would enhance the study's transparency.

6. In the methods, the coding process is described, but intercoder reliability is not mentioned. Without this, the rigor of qualitative coding might be questioned. It would be beneficial to report how consistency in coding was ensured, such as through independent coding by multiple researchers or the resolution of discrepancies.

7. In the results, the sample size is relatively small (N=18), and the ethnic distribution is skewed toward Chinese participants. However, this is not addressed as a potential limitation. Discussing the impact of this demographic distribution on the transferability of findings would add important context to the results.

8. In the discussion, the results are well interpreted, but there is no critical assessment of the potential unintended consequences of the intervention. It would be helpful to consider whether patient similarity-based risk communication could lead to increased anxiety, disengagement, or frustration among individuals with poor control.

9. In the discussion, the authors suggest that social comparison practices need to be stratified, but they do not propose a concrete framework for doing so. Providing a clearer direction on how future studies can operationalize the stratification of patient groups based on comparison tendencies would strengthen this section.

You may consider citing the following study:

DOI: 10.1371/journal.pone.0216563

This study involves expert consensus on structuring collaborative care models, which can be relevant when proposing a structured framework for stratifying patient similarity-based communication approaches.

(This is optional and should only be taken as a suggestion for the improvement of the manuscript.)

10. In the conclusion, it is stated that patient similarity-based risk communication is useful, but there is no critical assessment of who benefits the most from this approach. Clarifying which patient subgroups found this method most effective would make the conclusion more precise and informative.

You may consider citing the following study:

DOI: 10.3389/fpubh.2024.1323102

This study assesses the impact of pharmacist-led interventions on diabetes management, which can support discussions on which patient subgroups benefit most from specific intervention strategies.

(This is optional and should only be taken as a suggestion for the improvement of the manuscript.)

6. PLOS authors have the option to publish the peer review history of their article (what does this mean? ). If published, this will include your full peer review and any attached files.

**Do you want your identity to be public for this peer review?** For information about this choice, including consent withdrawal, please see our Privacy Policy .

Reviewer #1: **Yes: ** Mohamed Farouk Allam

Reviewer #2: No

Reviewer #3: No

---

## [Author Response · Author response to Decision Letter 1]

9 May 2025

Dear Editor and Reviewers,

Re: Response to reviewers’ comments on PONE-D-24-37711

Thank you for giving us the opportunity to improve our manuscript. We appreciate the time and effort you have taken to provide valuable feedback and have revised our manuscript.

We have provided our written responses to each of the points raised in your earlier letter.

EDITOR’S COMMENTS

1. Journal Requirements:

Response:

Thank you for highlighting this. The following changes have been made to the reference list in our revised manuscript:

a) Added new references #26, #27, #30, #31, #32, #35 and #42:

26. Sharafoddini A, Dubin JA, Lee J. Patient Similarity in Prediction Models Based on Health Data: A Scoping Review. JMIR Med Inform. 2017;5(1):e7. Published 2017 Mar 3. doi:10.2196/medinform.6730

27. Parimbelli E, Marini S, Sacchi L, Bellazzi R. Patient similarity for precision medicine: A systematic review. J Biomed Inform. 2018;83:87-96. doi:10.1016/j.jbi.2018.06.001

30. Ong R, Ng CJ, Gunasekaran K, et al. Utility of a patient similarity-based digital tool for risk communication to patients with type 2 diabetes mellitus: perspectives from primary care physicians in ambulatory care. PLoS One. 2025;20(3):e0319992. doi:10.1371/journal.pone.0319992

31. Beeney LJ, Fynes-Clinton EJ. The Language of Diabetes Complications: Communication and Framing of Risk Messages in North American and Australasian Diabetes-Specific Media. Clin Diabetes. 2019;37(2):116-123. doi:10.2337/cd18-0024

32. Lim WM. What is qualitative research? An overview and guidelines. Australasian Marketing Journal (AMJ). 2024;0(0). doi:10.1177/14413582241264619

35. Hennink M, Kaiser BN. Sample sizes for saturation in qualitative research: A systematic review of empirical tests. Soc Sci Med. 2022;292:114523. doi: 10.1016/j.socscimed.2021.114523. Epub 2021 Nov 2. PMID: 34785096.

42. Lipsey AF, Waterman AD, Wood EH, Balliet W. Evaluation of first-person storytelling on changing health-related attitudes, knowledge, behaviors, and outcomes: A scoping review. Patient Educ Couns. 2020;103(10):1922-34. doi:10.1016/j.pec.2020.04.014

b) Removed references #15-19 and #21

c) Renumbered the following references:

Previous # Renumbered as

#20 #23

#22 #24

#23 #25

#24 #19

#25 #20

#26 #21

#27 #22

#28 #18

#29 #15

#30 #16

#31 #17

#32 #28

#33 #29

#35 #36

#36 #37

#37 #38

#38 #39

#40 #41

#41 #33

#42 #43

#44 #45

#46 #47

#47 #48

d) References 1-14 and 34 remain unchanged

2. Additional Editor Comments:

The manuscript has a good deal of merit for publication after minor revisions.

Given the similarities between the two articles, we request the authors to provide a clear justification for the need to present these findings separately. Alternatively, they may consider combining their results into a single, comprehensive manuscript. This approach could strengthen the impact of their study by providing a more holistic perspective on the research question.

Response:

Thank you for your feedback. The previous study focused on primary care physicians who are the main stakeholders to communicate diabetes-related risks to patients using PERDICT.AI. The key themes from that study were ‘Education and motivation for subgroups of patients with T2DM’, ‘Patients who do not practise social comparison’, and ‘Potential for false reassurance or negative reactions from patients’. The findings informed us to narrow down the target recipients of the risk communication to patients with uncontrolled T2DM, instead of all T2DM patients regardless of glycaemic control. Therefore, this study explored the perspectives and experiences of this patient subgroup on how their diabetes-related risks were communicated to them by the physician using PERDICT.AI. The key themes were ‘I am myself’, ‘Motivation to be like the majority’, ‘Avoiding similar negative outcomes’, ‘Low risk does not equate to zero risk’, and ‘Motivation to replicate how others achieve positive outcomes’. Thus, the perspectives from these two major stakeholders are presented in two separate manuscripts.

We have added the following sentences (changes in bold) to the ‘Introduction’ section:

‘PERDICT.AI (Personalised Diabetes Counselling Tool using Artificial Intelligence), a digital tool based on an AI-driven patient similarity model, was developed by a team of primary care physicians (PCPs) and data scientists to support PCPs in patient similarity-based risk communication during their consultations with patients who have type 2 diabetes mellitus (T2DM) [28,29]. Based on the pilot study involving PCPs [30], risk communication leveraging on PERDICT.AI is likely to be more effective for patients with uncontrolled T2DM and who are receptive to social comparison. Furthermore, PCPs alluded to patients being discouraged or distressed when their glycaemic control is compared to that of other patients, particularly when their glycaemic control is due to circumstances out of their control. This is an unintended consequence of risk communication about diabetes complications that could interfere with diabetes self-management [31]. The findings informed us to narrow down the target recipients of the risk communication to patients with uncontrolled T2DM, instead of all T2DM patients regardless of glycaemic control. Therefore, this study explored the perspectives and experiences of this patient subgroup on how their diabetes-related risks were communicated to them by a PCP using PERDICT.AI. The findings will allow comparison of perspectives from these two major stakeholders on patient similarity-based risk communication using PERDICT.AI.’

The manuscript of the previous study with physician participants is cited as reference #30 in the revised manuscript. We have uploaded a copy named ‘journal.pone.0319992.pdf’, which replaces the old file ‘PONE-D-24-29527.pdf’.

References #30 and #31 are newly cited in the revised manuscript.

3. Comments from PLOS Editorial Office: We note that one or more reviewers has recommended that you cite specific previously published works. As always, we recommend that you please review and evaluate the requested works to determine whether they are relevant and should be cited. It is not a requirement to cite these works. We appreciate your attention to this request.

Response:

Thank you for highlighting this. We have read the articles recommended by the reviewers and have decided not to cite them. Do refer to our replies to the reviewer comments.

REVIEWERS’ COMMENTS

Reviewer #1:

1. Title: Appropriate and informative.

Response:

Thank you for your feedback.

2. Abstract: Appropriate and informative.

Number of words: 320 words (Maximum 300 words; not correct according to the instructions of the authors).

Response:

Thank you for your feedback. We have shortened the Abstract.

3. Keywords: Please use/add specific MeSH words like; Qualitative analysis, Type 2 diabetes mellitus, Singapore, PERDICT.AI.

Response:

Thank you for your feedback. We have revised the keywords to the following MeSH terms:

Diabetes Mellitus, Type 2

General Practice

Singapore

Digital Health

Qualitative Research

4. Introduction: Appropriate and informative.

Response:

Thank you for your feedback.

5. Objectives: Appropriate and informative.

Response:

Thank you for your feedback.

6. Methods:

The title of the section could be changed to be “Methodology” OR “Subjects and Methods” OR “Patients and Methods”.

Appropriate and informative.

Response:

Thank you for your feedback. We have renamed the section title to ‘Methodology’.

7. Statistical Analysis: Appropriate and informative; needs more details.

Response:

Thank you for your feedback. The 'Data analysis' section now reads as follows (changes in bold):

‘All audio recordings were transcribed verbatim and checked for completeness and accuracy. RO and KG familiarised themselves with the transcripts from the first two in-depth interviews and independently coded the transcripts. Open codes were assigned to the transcripts based on the study objective, and were combined to form a coding framework. RO and KG met to discuss and reach a consensus on shared meanings of the codes, and discrepancies were resolved through consultation with an experienced qualitative researcher CJN. RO coded the remaining transcripts and new codes were added iteratively after discussion with the study team. The data was managed using NVivo Windows Release 1.5.1. Emergent themes were identified. Thematic saturation was reached after 12 interviews. No new themes emerged from the remaining interviews.’

8. Results:

Text of the results: Appropriate and informative; but long.

Please do not repeat in the text what has been presented in the tables.

Response:

Thank you for your feedback. We have removed the repeated text, and also detailed the reasons for participant dropout (changes in bold) in the ‘Participant characteristics’ sub-section of the ‘Results’ section:

‘18 of the 20 participants in the intervention arm completed the patient similarity-based risk communication session and interview. Among the two participants who withdrew before the session, one cited loss of interest in the study and another cited ill health after unsuccessful rescheduling of the session. Characteristics of participants are detailed in Table 1.’

9. Discussion: Appropriate and informative, however it needs to discuss with more details every finding of the study.

Response:

Thank you for your feedback. We have added the following sentences and paragraphs (changes in bold) to the ‘Discussion’ session:

‘The study revealed diverse risk perceptions among the patient participants. Several of them were motivated to improve their glycaemic control to be compatible with most local patients with T2DM. In contrast, a few were resigned to be among the minority at risks of complications. The findings are consistent with PCPs’ perspectives from the previous study that risk communication using PERDICT.AI would more likely benefit patients who favour social comparison. While PCPs in the previous study had perceived that patients could potentially misinterpret or misunderstood their risks in using the PERDICT.AI [30], none of the patients in the present study alluded to such risk when they were compared with other “similar” patients.

With fear and risk aversity elicited among some participants, the study team recognises the potential for patient similarity-based risk communication to cause increased anxiety, frustration, and possibly disengagement in certain patients. This is consistent with the concerns PCPs raised in the previous study [30] and is a known unintended consequence of risk communication about diabetes complications that could interfere with diabetes self-management [31]. These occurrences can be minimised by PCPs’ awareness of their patients’ contextual factors contributing to glycaemic control, anticipation of negative reactions, and ability to frame and communicate risk information in a person-centric manner while taking the opportunity to motivate patients to improve their glycaemic control.

The findings are consistent with known literature on upward social comparison’s ability to engage competitiveness and facilitate information seeking to improve one’s health [18]. They are also consistent with known literature on risk perceptions having both “deliberative” (rule-based using absolute or comparative numerical information) and “affective” contextual factors that drive behavioural decisions [14,36-40], as demonstrated by participants who were motivated to reduce their complication risk to zero due to fear of being like the cohort minority who developed complications (i.e. “affective” risk perception), despite the minority being numerically small (i.e. not applying numerical principals seen in “deliberative” risk perception) [41].

Some participants wanted to know how to improve their glycaemic control and lower their risk of diabetes complications, seeking to replicate methods which worked for other patients. Future versions of PERDICT.AI should consider incorporating data on lifestyle measures. Patient narratives, through sharing of personal stories, can be used in tandem to motivate uptake of protective health behaviours and reduce fears among similar patients [42].

Patient similarity-based risk communication should be targeted at patients who are receptive to social comparison. Patients without such inclination may benefit from other modes of counselling to raise their risk awareness. Nuances in social comparison practices among the participants shows that self-reported information on social comparison tendencies alone do not reliably predict whether a patient would be receptive to patient similarity-based risk communication. Therefore, challenges are anticipated in identifying and stratifying the nuanced layers of social comparison practices among patients using existing questionnaires such as INCOM (Iowa-Netherlands Comparison Orientation Measure) [33]. Further research is needed to operationalise the stratification of T2DM patients based on their social comparison tendencies and to develop validated measurement tools for data collection. For example, a potential direction is to prospectively study social comparison occurrences, characteristics, and its consequences on T2DM patients practising behavioural change using ‘ecological momentary assessment’ after participants are trained to recognise day-to-day social comparison occurrences. This method intensely captures the nuances of time-sensitive within-person social comparison occurrences and effects over multiple short time periods throughout the day and is less prone to recall bias compared to retrospective self-reporting methods [43]. The findings would shed new insight on how to identify patients who are most receptive to patient similarity-based risk communication and to prioritise its use for these patients, with the aim of advancing their risk awareness and motivating the adoption of protective health behaviours [44]. The effectiveness of the risk communication would also depend on the PCP’s ability during a doctor-patient dialogue to frame the information in a person-centered manner, while avoiding the tendency to “climb probability trees” [45,46].

The study has limitations. Convenience sampling, rather than purposive sampling, was used due to challenges in recruiting patients into this study, introducing selection bias and limiting generalisability. Relevant attributes such as ethnicity and health literacy may not be adequately represented. Nevertheless, there was a good representation of the participants in terms of age, gender, education level, diabetes duration and control, which provided some degree of variation. The complication prevalence among the SPC should not be interpreted as an individual’s predicted risk of developing complications. For such predictions, validated risk calculators should be used, such as the UK Prospective Diabetes Study (UKPDS) [47] in the UK and Framingham Adult Treatment Panel ATP III [48] in Singapore for cardiovascular disease risk calculation.’

References #30, #31 and #42 are newly cited in the revised manuscript.

10. The Limitations Section of the study; more possible biases?

Response:

Thank you for highlighting this. We have described further limitations and biases (changes in bold) in the ‘Discussion’ section:

‘The study has limitations. Convenience sampling, rather than purposive sampling, was used due to challenges in recruiting patients into this study, introducing selection bias and limiting generalisability

---

## [Decision Letter · Decision Letter 1]

Patient perspectives on patient similarity-based risk communication for uncontrolled type 2 diabetes in primary care: A qualitative study

PONE-D-24-37711R1

Dear Dr.Ruiheng Ong,

We’re pleased to inform you that your manuscript has been judged scientifically suitable for publication and will be formally accepted for publication once it meets all outstanding technical requirements.

Kind regards,

Naeem Mubarak, PhD

Academic Editor

PLOS ONE

Additional Editor Comments (optional):

The authors have addressed all the comments. No further changes are required.

Reviewers' comments:

Reviewer's Responses to Questions

**Comments to the Author**

1. If the authors have adequately addressed your comments raised in a previous round of review and you feel that this manuscript is now acceptable for publication, you may indicate that here to bypass the “Comments to the Author” section, enter your conflict of interest statement in the “Confidential to Editor” section, and submit your "Accept" recommendation.

Reviewer #2: All comments have been addressed

Reviewer #3: All comments have been addressed

2. Is the manuscript technically sound, and do the data support the conclusions?

Reviewer #2: Yes

Reviewer #3: Yes

3. Has the statistical analysis been performed appropriately and rigorously? 

Reviewer #2: Yes

Reviewer #3: Yes

4. Have the authors made all data underlying the findings in their manuscript fully available?

Reviewer #2: Yes

Reviewer #3: Yes

5. Is the manuscript presented in an intelligible fashion and written in standard English?

Reviewer #2: Yes

Reviewer #3: Yes

6. Review Comments to the Author

Reviewer #2: The document has been reviewed significantly and is acceptable for publication. A few suggestions have been made in terms of language and can be found in the track changes mode in the document attached

Reviewer #3: (No Response)

7. PLOS authors have the option to publish the peer review history of their article (what does this mean? ). If published, this will include your full peer review and any attached files.

**Do you want your identity to be public for this peer review?** For information about this choice, including consent withdrawal, please see our Privacy Policy .

Reviewer #2: No

Reviewer #3: No

---

## [Editor Report · Acceptance letter]

PONE-D-24-37711R1

PLOS ONE

Dear Dr. Ong,

I'm pleased to inform you that your manuscript has been deemed suitable for publication in PLOS ONE. Congratulations! Your manuscript is now being handed over to our production team.

Kind regards,

on behalf of

Dr Naeem Mubarak

Academic Editor

PLOS ONE